# Amagmatic hydrothermal systems on Mars from radiogenic heat

Lujendra Ojha [1✉], Suniti Karunatillake [2], Saman Karimi [3] & Jacob Buffo[4]

Long-lived hydrothermal systems are prime targets for astrobiological exploration on Mars. Unlike magmatic or impact settings, radiogenic hydrothermal systems can survive for >100 million years because of the Ga half-lives of key radioactive elements (e.g., U, Th, and K), but remain unknown on Mars. Here, we use geochemistry, gravity, topography data, and numerical models to find potential radiogenic hydrothermal systems on Mars. We show that the Eridania region, which once contained a vast inland sea, possibly exceeding the combined volume of all other Martian surface water, could have readily hosted a radiogenic hydrothermal system. Thus, radiogenic hydrothermalism in Eridania could have sustained clement conditions for life far longer than most other habitable sites on Mars. Water radiolysis by radiogenic heat could have produced $H_2$, a key electron donor for microbial life. Furthermore, hydrothermal circulation may help explain the region's high crustal magnetic field and gravity anomaly.

[1] Department of Earth and Planetary Sciences. Rutgers, The State University of New Jersey, Piscataway, NJ, USA. [2] Department of Geology and Geophysics, Louisiana State University, Baton Rouge, LA, USA. [3] Department of Earth and Planetary Sciences, Johns Hopkins University, Baltimore, MD, USA. [4] Thayer School of Engineering, Dartmouth College, Hanover, NH, USA. ✉email: Luju.ojha@rutgers.edu

Hydrothermal systems are prime targets for astrobiological exploration as they contain redox gradients, energy sources, and aqueous solutions rich in elements essential for life, providing suitable conditions for sustained prebiotic syntheses[1]. On Earth, water circulation via hydrothermalism also causes significant cooling of the oceanic and continental lithosphere and plays a crucial role in regional metamorphism and ore formation[2–9]. Furthermore, the precipitation of magnetic minerals in hydrothermal environments can greatly influence crustal magnetism[10,11]. Hydrothermal systems may have played an equally important role in Mars' biological and geological history.

The two primary ingredients necessary for hydrothermal circulation, groundwater (even if of cryospheric provenance) and a heat source, were readily available on Mars during the Noachian eon [4.1–3.7 Ga ago]. A plethora of evidence on Mars suggests abundant surface water sustained by subsurface hydrology during the Noachian. These include spectroscopic detections of clays and other hydrous minerals[12], abundant valley networks in the mid-to-low latitudes[13], lakes[14], and putative shorelines[15–17], along with in situ observation of conglomerates[18] and possibly megafloods[19]. In addition, the spectral detection of various mineral phases commonly associated with hydrothermalism, such as Fe/Mg phyllosilicates, zeolites, prehnite, chlorite, serpentine, illite, kaolinite, carbonates, and hydrated silica[20–25] suggest that past aqueous alteration of the Martian crust occurred in the deep subsurface at high temperatures[26,27].

In the presence of groundwater, various heat sources can drive hydrothermal circulation on rocky planets. Those include magmatism related to plate tectonics and mantle convection[28], heat provided by impact events[29,30], serpentinization[31,32], and radioactivity[33]. Hydrothermal systems induced by some of those processes have been identified on Mars. Large impacts could have led to impact-induced hydrothermal systems, and several impact-induced hydrothermal systems have been proposed[34–36]. Chemically reduced impactors may induce both crustal and impact plume thermochemical pathways yielding $H_2$ and $CH_4$ as globally potent greenhouse gases[37,38]. The origin of some valley networks on Mars has also been associated with hydrothermalism via magmatic intrusion[39–41]. Silica-rich deposits identified in Nili Patera have been suggested to be the product of shallow hydrothermal systems linked to volcanism[42]. The draping of older clay by younger lava flows could have also initiated hydrothermal alteration in the eastern Nili Fossae region[23]. Serpentinization, in which highly reduced olivine and pyroxene-rich rocks react with water, liberates molecular $H_2$ and thermal energy, driving low-temperature hydrothermal systems. A similar process was likely active on early Mars[31], as indicated by the spectral detection of minerals associated with serpentinization across the southern highlands[21,26,43]. In most such cases, however, the hydrothermal systems are relatively short-lived, localized, or sporadic. Even local hydrothermalism at massive impacts like Hellas, and global warming from impact-induced $H_2$, may not have prolonged surface waters beyond tens of Ma time scales[30,37].

Radiogenic heat-driven hydrothermal systems are powered by the radioactive decay of long-lived heat-producing elements (HPE) with half-lives of billions of years (e.g., $^{238}U$, $^{235}U$, $^{232}Th$, and $^{40}K$). Thus, depending on their concentration, such systems can remain active for orders of magnitude longer (100–1000 Ma) than hydrothermal systems powered by alternative heat sources[33]. A prominent terrestrial example of a radiogenic heat-driven hydrothermal system is the Paleozoic Mt. Gee-Mt. Painter system in the Northern Flinders region of South Australia (MGPS). Radioactive decay within HPE-rich host rocks provides proximal heat energy for the MGPS, particularly the Mesoproterozoic granites emplaced by magmatic intrusions, some of which are extremely enriched in U and Th (2–3 times greater than expected

for Proterozoic crust)[44]. Radiogenic heat-driven hydrothermal systems may have been widespread on early Earth[45], as the heat production by radioactive elements would have been exponentially higher during the Hadean and Archean than the present[46,47]. However, the surface heat flow of the early Earth is a matter of considerable uncertainty[48]. Regardless, regions on Mars enriched in HPE during the Noachian could have sustained hydrothermal circulation for a prolonged period.

The Eridania basin on Mars comprises connected, small quasi-circular basins, which potentially originated as very ancient impacts[49] (Fig. 1). The mineralogy, geology, and the spatial context of the most ancient strata within Eridania suggest their formation in a deep-water hydrothermal setting[22]. The provenance of the liquid water in the Eridania basins is poorly known. Sparseness of valley networks in Eridania undermine surface runoff as the dominant water source[50]. Instead, the scarp-bounded benches in Gorgonum chaos[51], the concave hypsometry of Eridania basins[22,49], and the reducing deep-water environment indicated by $Fe^{2+}$-rich clay minerals are suggestive of Eridania being an ice-covered sea. The approximate size of the body of water in Eridania has been suggested to be $>10^6 km^2$, larger than the largest landlocked lake or sea on Earth (Caspian sea) by a factor of 3. The deep basins of Eridania contain >400-m-thick clay deposits, which, assuming terrestrial clay formation rate of 0.01–0.05 mm yr,$^{-1}$ would require tens of millions of years to form[52]. Thus, a heat source capable of sustaining a hydrothermal system for a prolonged period, such as radiogenic heat, may be necessary to explain the existence of Eridania. Michalski et al.[22] considered magmatic intrusion as a possible heat source that sustained the Eridania hydrothermal system. However, magmatic intrusions would be relatively short-lived compared to rocks rich in HPE sustaining the hydrothermal system in Eridania.

Here we show that Eridania and the surrounding regions have the highest abundance of Th, K, and cosmochemically equivalent U of any Noachian terrain. By combining chemical maps with geophysical data of Mars, we have shown that the region surrounding Eridania had one of the highest crustal heat flows on Mars during the Noachian[53]. Using this prior heat flow estimate, we show that the shallow subsurface temperature at Eridania would have easily exceeded the temperature threshold required for hydrothermal systems. In the Eridania region, several large impact basins have prominent Bouguer gravity anomalies suggesting a lack of viscoelastic relaxation of the Moho topography[54].

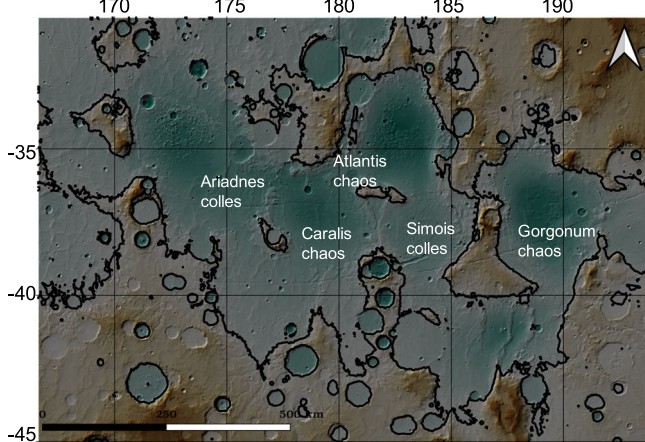

**Fig. 1 The topography of the Eridania basin.** Colorized Mola topography showing the topography of the Eridania basin and the surrounding region. The black contour lines bound the maximum (1100 m) level of an ancient sea.

We show that may be due to the crustal cooling by hydrothermal circulation, similar to the cooling of the oceanic and continental lithosphere by hydrothermal circulation on Earth[4,8,9]. The Eridania basin is located within the broader Terra Cimmeria/Terra Sirenum region, which has one of the highest crustal magnetic fields on Mars[55]. Such a high crustal magnetic field on Mars may also reflect augmentation of the primordial natural remnant magnetization by a hydrothermally induced chemical remnant magnetic field[56,57]. In summary, via synthesis of available geophysical and geochemical data of Mars along with numerical models of crater relaxation, we show that the Eridania region on Mars was potentially a long-lived radiogenic hydrothermal system that may have contributed to the region's enigmatic thermal and magnetic history besides providing a long-lived habitable environment.

## Results and discussion

**Th and K enriched regions on Mars**. To ascertain if the Eridania hydrothermal system was powered by radiogenic heat, we sought to find regions on Mars that are significantly enriched in Th and K. To that end, we used chemical maps of the Martian shallow subsurface derived from spectroscopy by the Mars Odyssey Gamma Ray Spectrometer Suite (GRS)[58]. We used an enhanced Student's $t$-test parameter, $t_i$, that measures the error-weighted deviation for each element from its bulk-average on Mars at each $5° \times 5°$ GRS grid (pixel) to find Th and K enriched regions (see "Methods"). Similar to our previous work[59,60], we define areas of significant enrichment as those with $t_i$ magnitude exceeding 1.5 (i.e., statistical confidence in directional deviation >94%). Within the Noachian highlands, the highest co-enrichment of Th and K on Mars is found in Eridania and its surrounding regions (Fig. 2). Our U chemical map, calculated using a cosmochemically constant Th/U mass ratio of 3.8[61], was used for heat flow calculation (Fig. 2). Furthermore, K and Th maps have the lowest spatial autocorrelation among all chemical maps as they are derived with only minimal spectral modeling from natural radioactivity instead of galactic cosmic particle flux-induced radioactivity. The reduced spatial autocorrelation translates to high spatial accuracy unlike for other elements, ensuring (cf., ref. [62]) that HPE enrichment is distinctive of Eridania in the Noachian terrain on Mars.

Outside Eridania, K and Th enrichment satisfying the enhanced Student's $t$-test with spatially significant extent is limited to the

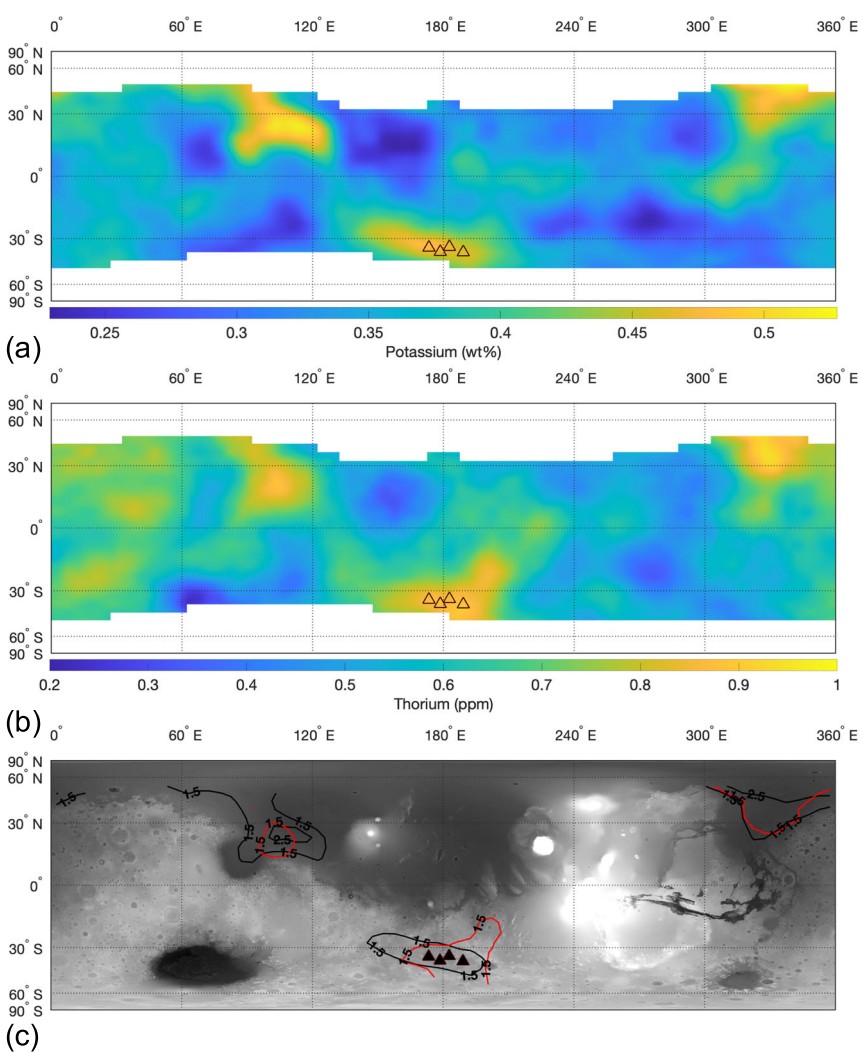

**Fig. 2 Distribution of the heat-producing elements in the shallow subsurface of Mars. a** Spatially smoothed potassium concentration in the shallow subsurface of Mars at $5° \times 5°$ in the mid-to-low latitudes, as derived from Mars Odyssey Gamma Ray Spectrometer suite. **b** Same as (**a**) but showing the concentration of Th. Rapidly increasing H abundance dilutes and increases numerical uncertainty for HPE concentrations in the polar latitudes. A mask has thus been applied to exclude such areas. **c** Contour map showing regions on Mars with significant enrichment of Th (in red) and K (in black). The contour labels correspond to enhanced Student's $t$-test parameter '$t_i$' that show regions with significant enrichment of Th and K compared to the bulk-average of Mars. The background is a shaded relief of Mars' topography and the black triangles show the location of Eridania basins.

northern lowlands, in the general area of the Vastitas Borealis Formation (VBF). Early works by the Mars Odyssey mission team[63,64] characterized that extensively, and found a preferential association with increasing abundance of Surface Type 2 (ST2) mineralogy[65]. Given a K/Th ratio corresponding to the crustal average, aqueous alteration (e.g., lowland water bodies) was excluded in preference for a volcaniclastic regolith derived from a mantle source with distinct large ion lithophile content[63,66]. Using impact exposures of the shallow subsurface, subsequent works corroborated further that spatial correspondence with ST2 suggests Hesperian aged volcaniclastics[67], unrelated to the Noachian basement indicated by quasi-circular "ghost" craters of the lowlands. Specifically, mapped Noachian geology is absent in the K–Th enriched lowlands, characterized instead by 20–40% areally of Hesperian volcaniclastics and up to ~10% Amazonian volcaniclastics [Fig. 6 by Karunatillake et al.[64]]. That contrasts dramatically with the Eridania enrichment where >60% of the area corresponds to Noachaian units [cf., Fig. 6 by Karunatillake et al.[64]]. Such association of the compositional signature with the younger upper regolith over the lowlands and VBF further separates the lowlands K–Th enrichment in time and provenance from Eridania's.

The enrichment of Th and K in the area surrounding Eridania may indicate aqueous alteration of HPE-rich magmatic intrusions by hydrothermal fluids, primordial mafic protolith rich in HPE, or subaqueous alteration in a submarine-like setting. The relative concentration of the two elements, placed in the context of overall geochemical trends in Eridania relative to the martian crust, helps resolve that uncertainty. To avoid analytical distributional bias, we use a non-parametric modified box-and-whisker (MBW) method developed expressly for planetary data[68,69], as shown in Fig. 3. For example, the observed Fe and Si depletions relative to the crust are consistent with a pervasiveness of residual rock from hydrothermal alteration, instead of precipitates from leached fluids[70,71]. Likewise, the exsolution of S-bearing phases such as $H_2S$ into the atmosphere during hydrothermalism would readily explain the absence of S enrichment in the area. The expectation that Al would resist leaching is realized with the presence of some high Al values in the area despite typically depleted values relative to the crust as shown in Fig. 3.

Hydrothermalism is known to leach K[72–74], decreasing the residual rock's K/Th, which we observe in Eridania (Fig. 3; Supplementary Fig. 1), albeit not at the level of anomalies found in regions like Memnonia Fossae (Supplementary Fig. 1)[75].

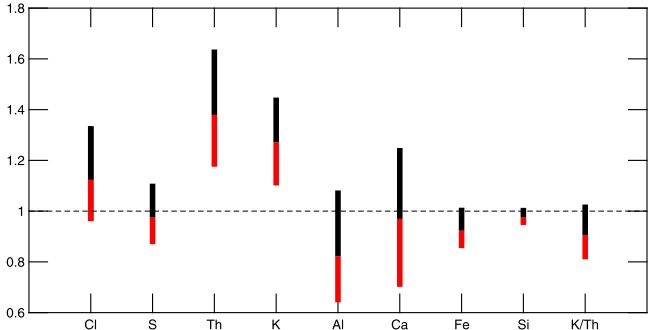

**Fig. 3 A Modified box-and-whiskers diagram to non-parametrically compare the distribution of major and minor elements, within the Th and K enriched region of Eridania, to the average Martian crust.** The distributional high within Eridania is represented by 75th percentile/25th percentile ratio at top of the upper box, the low by 25th/75th percentile ratio by the bottom of the lower box and the typical by the ratio of medians as the boundary between them. Deviation from unity, even at a small fraction, highlights instances of notably distinct regional chemistry relative to the crust, such as for Fe, Si, K, Th, and K/Th.

Hydrothermalism may also be responsible for the lower than average K/Th ratio in some other regions of Mars (Supplementary Fig. 1). If clays on Mars primarily formed via hydrothermalism[27], regions with clay-bearing units should exhibit lower than average K/Th ratio. The spatial distribution of clay-bearing units and K/Th ratio on Mars show a potential correlation (Supplementary Fig. 1), however, given the dramatic contrast in lateral resolution between nuclear and VNIR spectroscopy and lack of knowledge about the lateral extent of the mineral outcrops, any putative correlation must be treated with caution. Regardless, the low K/Th ratio in Eridania along with previous geomorphic and spectral evidence for an inland sea is suggestive of leaching of K by hydrothermal fluids.

The K-Th-enriched area in Terra Cimmeria and Terra Sirenum region is also much larger than the Eridania basins (Fig. 2); thus, subaqueous or submarine alteration alone cannot explain the enrichment of Th and K in this whole region. While the regional context from our limited suite of chemical maps cannot eliminate aqueous alteration as a contributor to K and Th enrichment in Eridania, the general geochemical consistency with hydrothermalism is reasonably attributable to the interaction of hydrothermal fluids with HPE enriched rocks at depth. Regardless of whether the observed enrichment of HPE in this region indicates aqueous alteration of HPE-rich intrusions by hydrothermal fluids or a primordial mafic protolith rich in HPE, it suggests that radiogenic heat played a significant role in the Eridania hydrothermal system. The enrichment of K and Th could have also led to the production of $H_2$ by water radiolysis[76,77]. Several microbial communities on Earth rely on $H_2$ as their primary electron donor[78,79], and studies have suggested that radiolysis could have supported possible Martian life[80,81].

**Can the observed enrichment of Th and K in Eridania drive hydrothermal circulation?** Given their incompatible nature, a strong fractionation of HPE in the Martian crust is an inevitable consequence of mantle magmatism. Previously it has been suggested that during the early differentiation of Mars, ~50–70% of the bulk HPE was partitioned into the crust[82,83]. A notable effect of a strong partitioning of the HPE into the crust is that the resulting high crustal heat flow can lead to hydrothermal circulation. For example, in MGPS, hydrothermal activity has continued for the last ~300 Ma due to the high geothermal gradients caused by the radiogenic basement, with mean heat production in the crust estimated to be $9.9 \times 10^{-6}$ W m$^{-3}$ [84]. The present-day surface heat flow in MGPS is estimated to be ~85 mW m$^{-2}$ with crustal heat sources contributing over 70 mW m$^{-2}$ and mantle heat flow only contributing 10–15 mW m$^{-2}$ [84]. The longevity of this hydrothermal system is demonstrated by the small-scale hydrothermal activity that persists to this day (300 Ma later) in Paralana Hot Springs where water emerges at temperatures exceeding 65 °C[85].

To assess if the observed concentration of K, Th, and U in Eridania and the surrounding region provided sufficient radiogenic heat for a hydrothermal system, we estimate plausible Noachian geotherms using heat flow estimates from our previous work[53]. Crustal heat flow estimates depend on the heat production rate of the crust, crustal thickness, and crustal density (see "Methods"). As in prior works, going back to those by Hahn et al. (2007, 2011), we approximate the bulk regolith chemistry to be representative of the crust[61,86]. While that introduces a conceptual uncertainty as the age or provenance of the bulk regolith relative to that of the underlying crust is not well known, prior works show that regolith chemistry can serve as an informative proxy for crustal processes[83,87]. That is notably true when mapped geology overlaps with distinct geochemistry[60,69,88] as for Eridania, albeit with greater uncertainty in areas of thick mantles of fine siliciclastics (i.e., dust)[89]. Further, ejecta from

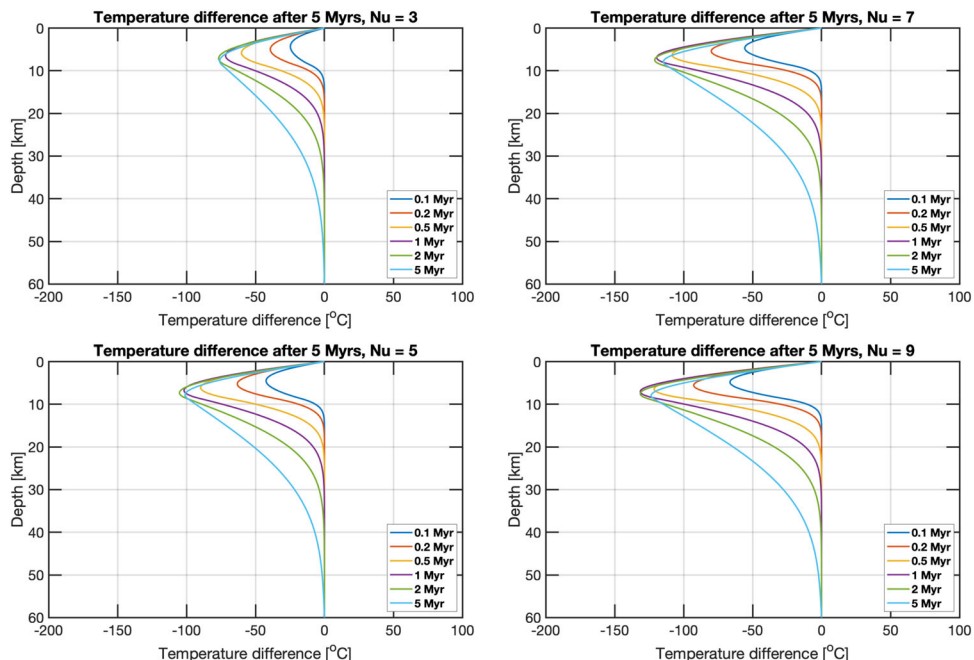

**Fig. 4 Temperature difference as a function of depth relative to the purely conductive thermal profile for simulations of different Nusselt numbers (Nu).** The four panels show the difference in temperature between a purely conductive thermal profile and thermal profiles due to hydrothermal circulation. The vigor of the hydrothermal circulation is given by the Nusselt number (Nu). The different colored lines show the temperature profile at various timescales up to 5 million years.

Hellas and Argyre are expected to have only thinly mantled the highlands of Terra Cimmeria and Terra Sirenum; thus, the Eridania surrounding region may represent the pre-Hellas surface better than most other regions on Mars[90]. Eridania's generally low dust cover increases the likelihood that the observed HPE enrichment is representative of the underlying crust.

With the preceding simplification, we estimated the crustal heat flow using GRS-derived chemical mass fraction maps of K and Th, gravity-derived crustal thickness models, and estimate of crustal density from representative Martian meteorites[53]. The crustal heat flow alone in Eridania and the surrounding region could have exceeded 45 mW m$^{-2}$ during the Noachian (Supplementary Fig. 2). The crustal correlation between K and Th[74], when applied to the currently depleted K/Th ratio (Fig. 3), may even imply ~10% higher K abundance in the parent rock than currently observed and correspondingly higher crustal heat flow at the start of hydrothermalism. Even with a modest mantle heat flow contribution of 20 mW m$^{-2}$, the surface heat flow in this region could have exceeded 65 mW m$^{-2}$ [53], consistent with previous Noachian heat flow estimates from crater relaxation models[91], lithospheric flexure models[92], and numerical thermal models[93].

The various hydrothermal phases identified in the Eridania region, such as saponite, talc-saponite, Fe-rich mica, Fe- and Mg-serpentine, Mg-Fe-Ca carbonate, and probable Fe-sulfide[22] implicate a formation environment in a deep hydrothermal setting where fluids with elevated temperature interacted with the host rocks[52]. For example, serpentines form by hydrothermal alteration of ultramafic rocks at temperatures ranging from ambient to 400 °C[94]. Nontronite formation on Mars requires elevated temperature in the range 20–40 °C[52]. A steady-state geotherm with our heat flow estimates and a mean crustal conductivity of 3 W m$^{-1}$ K$^{-1}$ show that these temperature values could have been possible at shallow depths in Eridania (Supplementary Fig. 3). Thus, the observed K, Th, and computed U abundance in the Eridania region could have readily sustained a radiogenic heat-driven hydrothermal system.

**Evidence for hydrothermal circulation from the lack of viscoelastic relaxation of craters.** On Earth, hydrothermal circulation plays a significant role in cooling the young oceanic and continental lithosphere[4,8,9]. Thus, if hydrothermal circulation was active in Eridania, it may have similarly contributed to the cooling of the lithosphere. The depth of hydrothermal circulation depends on the permeability profile of the crust of the planet. Specifically, the groundwater's penetration depth is limited to the brittle-ductile transition (BDT) depth, below which the permeability is too low for fluids to advect heat[6]. In the young oceanic lithosphere and continental crust of Earth, the groundwater circulation depth can extend to 10 km[95,96]. Due to reduced gravity, the confining pressure at 10 km on Earth would be reached at a depth of ~25 km in the Martian crust[97] so crustal cooling could extend to great depths on Mars (Supplementary Fig. 4). However, the lower gravity will also reduce the Rayleigh number (a measure for determining the buoyancy-driven fluid flow), thereby reducing the hydrothermal activity's vigor. The presence of other driving forces, such as overpressure from a thick ice cover over a sea as hypothesized for Eridania[22], can counteract the effect of the reduced gravity.

We used 1D thermal evolution models[8] to investigate the efficacy of hydrothermal circulation in lowering the crustal temperature profile on Mars. The magnitude of the crustal cooling is largely dependent on the dimensionless Nusselt number (Nu), comparing the relative importance of the total heat flux to the conductive heat flux. In the absence of hydrothermal heat transport, Nu = 1, and the total heat flux is identical to the conductive heat flux. When Nu >1, the total heat flux has contributions from the hydrothermal circulation of water. To test a broad range of Nu, we vary its value from 3 (minor hydrothermal circulation) to 9 (vigorous hydrothermal circulation). For Nu = 9 over a 5 Ma period, we find that hydrothermal circulation on Mars can extend to tens of km depth, lowering the shallow crustal temperature profiles by more than 100 K (Fig. 4), still many factors above the freezing point of water.

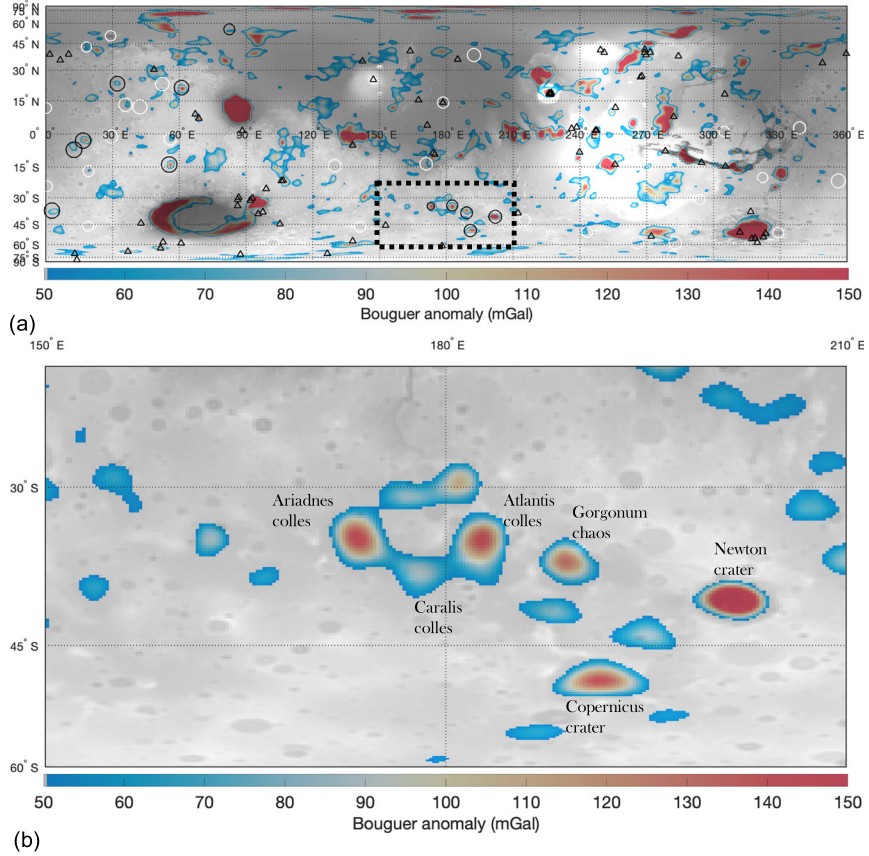

**Fig. 5 Filter Bouguer gravity anomaly map of Mars. a** A filtered Bouguer gravity anomaly map of Mars overlaid on top of the Mars Orbiter Laser Altimeter (MOLA) elevation map shaded relief. A bandpass filter with a lower and upper limit of 10 and 90 was used to filter out gravity signatures associated with spherical harmonic degree less than 10 and higher than 90. The circles show impact craters on Mars with a diameter between 200 and 900 km. Impact craters with associated Bouguer anomaly are shown in black while white circles show impact craters with no prominent Bouguer anomaly. Note that the colormap is scaled such that smaller Bouguer anomalies (<50 mGal) are not displayed for clarity. The triangles show all identified volcanic structures on Mars. **b** Bouguer gravity anomaly map of Eridania and the surrounding region. The location of Eridania basins and nearby impact basins are annotated in the figure.

A notable effect of the crustal cooling by groundwater circulation is the inhibition of the viscoelastic relaxation of the surface and Moho topography[9,98]. This is because viscoelastic relaxation depends on the material viscosity, which is modulated mainly by the materials' temperature and rheology. Previously, hydrothermal cooling models have been proposed to help explain the preservation of ancient crustal thickness variation on Mars[98]. On Mars, almost all basins between 275 and 1000 km in diameter show very shallow depths and limited crustal thinning for their size, indicating that viscoelastic relaxation was a dominant geological process[54] that reduced mantle relief via lower crustal flow. However, Eridania basins and impact craters in the vicinity, including Newton and Copernicus, have large Bouguer gravity anomalies (Fig. 5), suggestive of Moho topography that did not undergo significant relaxation. In fact, of the 40 impact craters on Mars with a diameter between 200 and 500 km (Supplementary Table 1), only ~12 impact craters have prominent Bouguer gravity anomalies (Supplementary Table 1), with five of them located in Eridania and the surrounding region (Fig. 5). Newton crater, in the vicinity of the Eridania region, has the highest Bouguer gravity anomaly of any known impact basin on Mars with a diameter <500 km (Fig. 4; Supplementary Fig. 5). Despite its late Noachian age (similar to Eridania), Newton is also remarkably well preserved and is one of the deepest craters in the Noachian highland (Fig. 6). The other craters with a comparable depth to Newton are much younger, Galle and Lyot craters

(Fig. 6), and both are devoid of prominent Bouguer gravity anomalies. The lack of viscoelastic relaxation at Newton crater led ref. [54] to conclude that locally enhanced hydrothermal cooling of the sort proposed by Parmentier and Zuber[98] may have significantly reduced crustal temperatures in this area. The lack of surface and Moho relaxation of Newton (impact age[91]: 3.95–4.00 Ga), Copernicus (impact age[91]: 3.91–3.97), and Eridania basins (impact age[22]: >4 Ga) provides further corroboration that this region experienced significant crustal cooling for an extended period.

If the Eridania hydrothermal system were primarily driven by the heat from magmatic intrusion at great depths[22] instead of heat from radioactivity in the crust, then the high basal heat flow should have induced significant viscoelastic relaxation of the surface and Moho topography in this region. While both heat sources can elevate the temperature in the crust and drive hydrothermal circulation, radioactivity within the crust does not notably change the temperature structure at depths (Supplementary Fig. 3). In contrast, the temperature at depth is significantly augmented in the presence of a magmatic heat source (Supplementary Fig. 3). Thus, if the hydrothermal system in Eridania was driven by magmatic heat at depth, we should expect the basins to have undergone viscoelastic relaxation. To quantitatively investigate this, we used the commercially available Marc-Mentat finite element package (http://www.mscsoftware.com) to model the viscoelastic relaxation (or lack of) of Newton

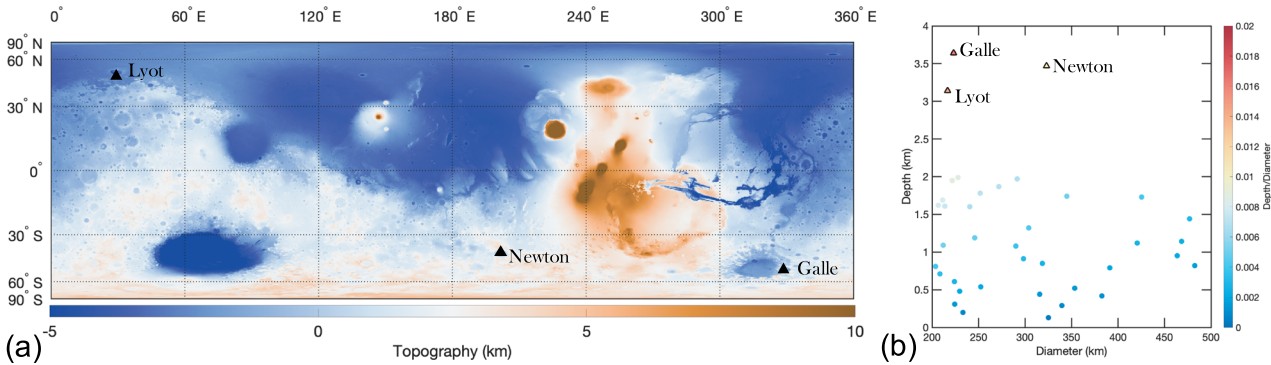

**Fig. 6 Depth to diameter ratio of impact craters on Mars. a** Topography map of Mars derived from the Mars Orbiter Laster Altimeter data showing the deepest craters on Mars with diameters between 200 and 500 km. **b** Depth to diameter ratio of all impact craters on Mars between 200 and 500 km in diameter.

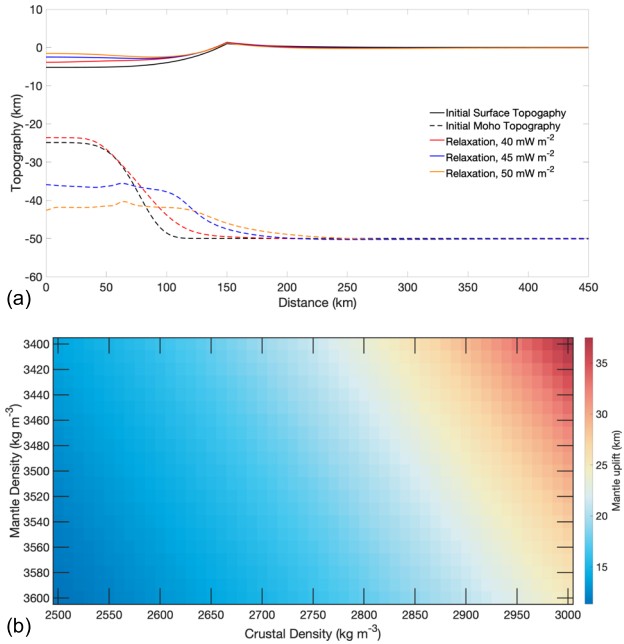

**Fig. 7 Viscoelastic relaxation models of the surface and the Moho topography of Newton crater. a** The solid and dashed black lines show the initial simulated surface and Moho topography. The colored lines show the surface and Moho topography due to viscous relaxation for various heat flow values after 100 Ma. **b** Mantle uplift necessary to achieve isostasy in Newton crater as a function of crustal and mantle density.

crater post-formation under various thermal and mechanical conditions. Similar to our previous work, we employ finite element analysis with non-linear viscoelastic rheology to simulate the deformation of impact craters at the surface and within the subsurface[91]. Under hydrous rheology, we find that if the basal heat flux were higher than 40 mW m$^{-2}$, then both the surface and the Moho topography of Newton should have undergone significant relaxation, consistent with previous findings[54] (Fig. 7). In contrast, a basal heat flux of 40 mW m$^{-2}$ or lower, by inhibiting Moho relaxation, would have led to Newton turning into a mascon basin[99]. However, none of the basins considered here Newton, Copernicus, or Eridania basins are mascons, since they lack a positive free-air gravity anomaly (Supplementary Fig. 6). Further, Newton is one of the deepest craters on Mars

with a prominent Bouguer anomaly suggesting that neither the surface nor the Moho topography of Newton underwent any significant relaxation. The presence of prominent Bouguer anomaly in the Eridania basin and Copernicus crater further suggests that regional magmatic intrusion likely did not play a significant role in the hydrothermal circulation in Eridania. Alternatively, the preservation of ancient gravity anomalies at Newton and other basins of Eridania could also be due to a heat-pipe-driven process which operates on a global scale and is postulated to produce a thick, cold, and strong lithosphere very early in Mars' history[100]. However, that should have led to the widespread preservation of the Moho topography of impact basins, which is not observed (Fig. 5).

Deep hydrothermal circulation and subsequent precipitation of single-domain magnetite could also have been a possible magnetization source in the Martian crust[56,97,101,102]. It is generally assumed that most of the observed crustal natural remnant magnetization (NRM) on Mars was acquired via thermal remnant magnetization (TRM) during the cooling of the crust. However, chemical remnant magnetization (CRM) via precipitation of single-domain magnetite in hydrothermal settings can significantly augment the primordial TRM, provided that the dynamo was still active during the time of the hydrothermal circulation[103]. The larger areas of Terra Sirenum (TS) and Terra Cimmeria (TC) surrounding the Eridania basin have the highest magnitude of the crustal magnetic field on Mars (Fig. 8). The same region of TS/TC is also significantly enriched in Th and K. In fact, the TC/TS regions are the most Th and K enriched region in Noachian highlands of Mars and, correspondingly, the region on Mars with a high possibility of vigorous and long-lived hydrothermal reactions.

Radiogenic heat-driven hydrothermal systems are long-lived and could be prime targets for astrobiological exploration on Mars. Based on the enrichment of HPE, such as Th, K, and U, we show that Eridania may have been the likeliest radiogenic heat-driven hydrothermal system on Mars. The lack of surface and Moho viscoelastic relaxation in the Eridania surrounding region strongly corroborates our amagmatic radiogenic heat-driven hydrothermal system hypothesis. In addition to potentially providing a long-lived habitable environment, hydrothermal circulation in this area could have contributed notably to the observed crustal magnetic field. Furthermore, as a site that hosted a large volume of liquid water, radiolytic reactions may have sustained energetics and chemical redox gradients conducive to the appearance of life[104], while associated by products, like H$_2$[38], would have affected the global climate.

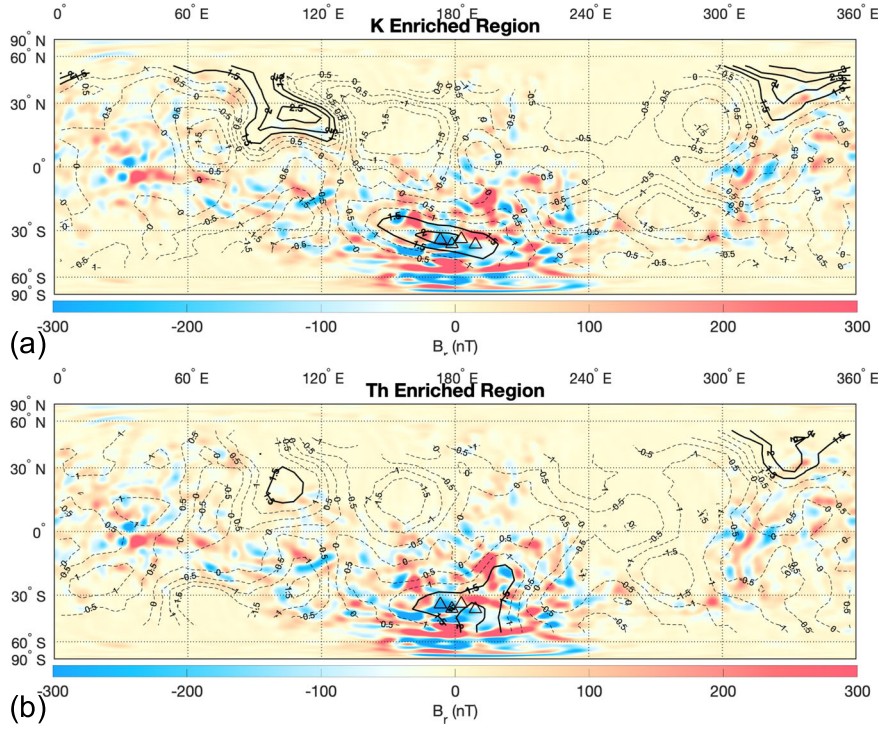

**Fig. 8 Map of the radial magnetic field of Mars and correlation with elemental K, Th, and Fe. a** K enriched regions on Mars in color overlaid on MOLA shaded relief. The bold black lines show areas on Mars that are statistically enriched in K. The dashed black lines are areas on Mars that have the average K composition or concentration less than the average K composition. **b** Same as (**a**) but for the elemental composition of Th.

## Methods

**Chemical maps and Th, K enriched regions on Mars**. We derive the regional shallow subsurface composition of the Martian crust to decimeter depths with GRS data. GRS measures the spectrum of gamma photons emitted from the Martian surface; characteristic spectral peaks from specific nuclear reactions allow the quantification of several major rock-forming elements, along with select minor and trace elements (Al, Ca, Cl, Fe, H, K, S, Si, Th)[58]. Peak area above the continuum can be used to infer the percentage mass fraction (wt%) of each element over an area of the planet's surface, leading to chemical abundance maps. Due to the nature of the derivation methods (e.g., elements like Fe and Si, with abundances determined from scatter and capture nuclear reactions from the galactic cosmic flux), the chemical maps are generally restricted to the mid-to-low latitudes (roughly $+/-50°$ latitude) where H abundances are sufficiently low. While K and Th are naturally radioactive, we also consider those within the midlatitudes to avoid mass dilution effects from H and to enable direct comparisons with the remaining elements.

We sought to find Th, K enriched regions on Mars. Similar to our previous work, we used an enhanced Student's $t$-test parameter, $t_i$, that measures the error-weighted deviation for each element from its bulk-average[59,64]:

$$t_i = \frac{c_i - m}{\sqrt{s_{m,i}^2 + s^2}};$$ (1)

where $c_i$ is the wt% of an element, $m$ is the global arithmetic mean wt%, $s_{m,i}$ is the numerical uncertainty of $c_i$, and $s$ is the standard deviation of the data.

**Crustal heat flow estimates**. The heat production rate of a crust ($Q_c$) is given by:

$$Q_c = \left[0.9928 C_U H_{238U} \exp\left(\frac{t\ln2}{\tau^{\frac{1}{2}}238U}\right) + 0.0071 C_U H_{235U} \exp\left(\frac{t\ln2}{\tau^{\frac{1}{2}}235U}\right) \right.$$
$$\left. + C_{Th} H_{232Th} \exp\left(\frac{t\ln2}{\tau^{\frac{1}{2}}232Th}\right) + 1.191 \times 10^{-4} C_K H_{40K} \exp\left(\frac{t\ln2}{\tau^{\frac{1}{2}}40K}\right)\right]$$ (2)

Here, $C$ and $H$ represent the concentration and heat release constants of the radiogenic elements, $t$ is time, and $\tau^{1/2}$ are the half-lives of the radioactive elements[105]. The concentration of the heat-producing elements is estimated using the GRS chemical maps. Heat release constants and the half-lives of the radiogenic elements are provided in numerous previous work[61,105]. The heat production rate is multiplied by the estimate of crustal density and crustal thickness to derive a first-order estimate of Noachian crustal heat flow. We use gravity-derived crustal thickness models which estimatethe average Noachian crust to be >50 km. An average crustal density of 2900 kg m$^{-3}$ is assumed for this work. The effect of these parameters on the average crustal heat flow is discussed in our previous work[53].

**Gravity analysis**. The spherical harmonic representation of the gravitational potential ($U$) exterior to a planet's surface is given by:

$$U(\theta, \Phi) = \frac{GM}{r} \sum_{l=0}^{\infty} \sum_{m=-l}^{l} \left(\frac{R_0}{r}\right)^l C_{lm} Y_{lm}(\theta, \Phi);$$ (3)

where $G$ is the gravitational constant, $M$ is the mass of the planet, $Y_{lm}$ is the spherical harmonic function of degree $l$ and $m$, $C_{lm}$ represents the spherical harmonic coefficient of the gravitational potential at a reference radius $R_0$, and with colatitude and longitude ($\theta$, $\Phi$). In this work, we use the most recent gravity model of Mars (GMM-3[106]) expanded to degree and order 120 (spatial resolution ~177 km). The average uncertainty in the gravity coefficients nearly equals the coefficient magnitude around degree 95; thus, we limit the expansion of the gravity field to up to degree 95.

We calculate the Bouguer anomaly from the surface topography and free-air gravity. We used the Mars Orbiter Laser Altimeter (MOLA) topography for the Bouguer correction. Although the resolution of the Martian topography supersedes the gravity resolution, we limit the expansion of the topography to degree and order 95, similar to the gravity field's resolution. A finite-amplitude correction of the 7th order is used for the Bouguer correction. Lower degrees of the Bouguer gravity field are sensitive to deep structures, including heterogeneities in the mantle density[107]. The goal of our gravity analysis is to examine and analyze the Bouguer anomaly arising from mantle uplift in impact craters, the source of which could be much shallower. Therefore, we apply a bandpass filter (using a Hanning window) with the lower and upper limit set to spherical harmonics degree 10 to 90. This allows us to examine the gravity anomalies related to impact processes more clearly while filtering out density anomalies that may arise from density variations within the mantle or deeper in the crust.

**Magnetic analysis**. The spherical harmonic representation of magnetic potential $V$ is given by

$$V = a \sum_{l=1}^{\infty} \left(\frac{a}{r}\right)^{l+1} \sum_{m=0}^{l} [g_{lm}\cos m\Phi + h_{lm}\sin m\Phi] P_{lm}(\cos\theta);$$ (4)

where $P_{lm}$ are the Schmidt-normalized Legendre functions, while $g_{lm}$ and $h_{lm}$ are the Gauss coefficients with colatitude ($\theta$) and longitude ($\Phi$). Here we use the latest scalar potential field model which combines the magnetic field data sets collected by two different spacecrafts: Mars Global Surveyor (MGS) magnetometer and Mars Atmosphere and Volatile Evolution (MAVEN) magnetometer over 13 cumulative years[108]. The model is expanded to degree and order 134, corresponding to a spatial resolution of ~160 km.

**Crater relaxation models**. We use the commercially available Marc-Mentat finite element package and numerically model Newton crater's post-impact evolution using viscoelastic rheology. Each relaxation simulation set consists of three steps: (i) building a finite element mesh, (ii) running a thermal simulation, and (iii) running a mechanical simulation. In our simulations, we use a two-layer axisymmetric mesh of one radial plane with the domain size of $3R \times 3R$ ($R$ = radius) to ensure that far-edge boundary effects do not affect our results. Given the diameter of the Newton basin (~300 km), we use a planar mesh instead of a spherical mesh. The density values for the crust and mantle are set to 2900 and 3500 kg m$^{-3}$, respectively. The crustal thickness in our simulation is set to 50 km, equal to that of the average crustal thickness of Mars[109]. The rate of relaxation is primarily dependent on the mantle heat flow, so reasonable variations in the density and thickness of the crust do not significantly impact our results. The crater depression shape is modeled with a fourth-order polynomial, while the ejecta blanket is modeled with an inverse third power law[110]. We use the depth-diameter relation of ref. [111] to estimate the initial depth and rim height of the crater (5300 m and 900 m, respectively)[111]. After an impact event, the mantle beneath the crater depression is uplifted to the isostatic equilibrium level[112], the amplitude ($h$) of which can be approximated by a simple mass balance equation:

$$h = \frac{\rho_c}{(\rho_m - \rho_c)} z; \tag{5}$$

where $\rho_c$ and $\rho_m$ are the densities of the crust and mantle, and $z$ is the depth of the crater. Assuming a crustal density of 2900 kg m$^{-3}$, a mantle density of 3500 kg m$^{-3}$, Newton crater's current depth of 3500 m, the mantle uplift likely exceeded 20 km at the time of the formation of this basin. We model the shape of the mantle to resemble a gaussian-like function similar to our previous work[91]. The total number of elements in the mesh exceeds $10^4$.

The viscoelastic deformation, or lack thereof, of an impact basin, is strongly controlled by the planetary body's thermal structure in the uppermost layer. We set the surface temperature to 210 K and create various thermal profiles assuming various background heat flow values. We also apply a boundary condition that approximates the effects of the impact heat (see Karimi et al.[91] for complete details). One of the parameters we seek to constrain in this work is the heat flow value in Noachian Mars, so we vary the heat flow estimates between 40 and 60 mW m$^{-2}$. We set the mantle and the crustal thermal conductivity to 4 and 2.5 W m$^{-1}$ K$^{-1}$ [113]. The output from the thermal simulation is input into the mechanical simulation.

In the mechanical simulation, the crust and mantle are assigned appropriate rheological properties, including elastic parameters. The elastic Young's moduli for the crust and mantle are set to 60 and 120 GPa, respectively, while the Poisson's ratio for both is set to 0.25[105]. The flow law parameters of hydrous basalt and peridotite are applied for the crust and mantle in the mechanical simulations[114,115]. The following boundary conditions are further applied: free-slip boundary conditions for the two sides of the mesh and fixating of the nodes at the bottom of the mesh. The Martian gravity is the driving force behind relaxation, and thus a gravitational acceleration of 3.7 m s$^{-2}$ is applied for the entire mesh in the vertical direction. The minimum viscosity in the simulations is set to $10^{21}$ Pa s, which keeps the computational time tractable[116]. The simulations are run for 100 Ma as previous work has shown 100 Ma to be sufficient to capture any considerable relaxation of large crater[117].

**Hydrothermal cooling models**. The depth to which groundwater can percolate and enable hydrothermal circulation depends on the permeability profile of the crust. Permeability decreases with depth due to the closure of pore spaces. The permeability profile of the Martian crust is not known, so in this study, we adopt the depth-dependent permeability of Saar and Manga, which was constrained by hydrological, thermal, seismic, and modeling studies in the Oregon Cascades. The permeability profile was adapted for Mars by scaling the gravity to match that of Mars (Supplementary Fig. 4). The permeability profile of the Martian crust may vary from the profiles we have adopted in this work; however, the goal here is to only investigate the first-order effect of crustal cooling by hydrothermalism on Mars. The thermal cooling of the Martian crust was modeled by a previously published 1D thermal conduction model[8].

## Data availability

All data needed to evaluate the conclusions of the paper are present in the paper, Supplementary materials, or through NASA's Planetary Data System (PDS). The Mars Odyssey Gamma Ray Spectrometer (GRS) derived chemical maps were derived from the spectral data archived at the PDS (https://pds-geosciences.wustl.edu/missions/odyssey/grs.html) by applying the spectra-to-chemistry modeling methods described in previous work[58,65,118]. The derived chemical maps have also been described and analyzed in other recent works[59,119]. The gravity model of Mars is available through the gravity model page of PDS node (https://pds-geosciences.wustl.edu/dataserv/gravity_models.htm). The crater database used to create Figs. 5 and 6 is from previously published work (https://doi.org/10.1029/2011JE003967) and (https://doi.org/10.1029/2011JE003966). The spherical harmonic model of the Martian magnetic field model used to create Fig. 8 can be accessed from previously published work (https://doi.org/10.1029/2018JE005854). The distribution of phyllosilicates shown in Supplementary Fig. 1 is a compilation of several publications and can be found here (https://doi.org/10.1146/annurev-earth-060313-055024).

## Code availability

The crater relaxation models were created using commercial software called Marc-Mentat (http://www.mscsoftware.com). The codes to analyze gravity and magnetics data can be found here (https://www.mathworks.com/matlabcentral/fileexchange/71416-slepian_alpha). No other custom algorithm was generated as a part of this work.

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

## Acknowledgements

This work was funded by a startup grant to L.O. by Rutgers University. S.K.'s participation was funded by NASA-MDAP grant 80NSSC18K1375.

## Author contributions

L.O. conceived the project and performed the geochemical, gravitational, and heat flow analysis. S. Karunatillake aided in geochemical and statistical interpretation of the results. S. Karimi contributed to the crater relaxation models. J.B. contributed to the heat flow modeling and interpretation of the results. L.O. wrote the paper with significant feedback from all other authors.

## Competing interests

The authors declare no competing interests.
