## [Peer Review File · Nature Communications]

Reviewers' Comments:

Reviewer #1:

Remarks to the Author:

Review of "Amagmatic hydrothermal systems on Mars from radiogenic heat" by Ojha et al.

This is a very nicely written paper that presents a study of the possibility of radiogenic heating causing hydrothermal systems to operate in the Noachian of Mars based on results from the Odyssey GRS data and focusing on the Eridania region.

Overall, I believe this is a very nice paper that could definitely be published in Nature Comms with some minor edits, mostly grammar related, which I will now go over.

First, the minor scientific questions that could be dealt with quite quickly:

1. On Figure 2-6 and S1, I would recommend putting a label on the figure showing what the triangles stand for. I couldn't find this discussed in the text. The scale bars should be shorter for clarity and the longitude should be at the top of the figure, not the middle.

2. On line 40, the authors talk about hydrothermal activity influencing crustal magmatism, and they reference 2 articles, I would suggest they also reference this article:

Lillis, Robert J., Josef Dufek, Walter S. Kiefer, Benjamin A. Black, Michael Manga, Jacob A. Richardson, and Jacob E. Bleacher. "The Syrtis Major Volcano, Mars: A Multidisciplinary Approach to Interpreting Its Magmatic Evolution and Structural Development." *Journal of Geophysical Research: Planets* 120, no. 9 (September 1, 2015): 2014JE004774. <https://doi.org/10.1002/2014JE004774>.

3. On line 50, the authors talk about the formation of minerals associated with hydrothermal activity, but did not mention carbonate as a possible serpentinization product, they probably should reference this study which looks at that possibility in Nili Fossae:

Brown, Adrian J., Christina E. Viviano, and Timothy A. Gouge. "Olivine-Carbonate Mineralogy of the Jezero Crater Region." *Journal of Geophysical Research: Planets* 125 (2020): <https://doi.org/10.1029/2019JE006011>.

4. On line 98, the authors reference Ojha et al (in press), I think it would be good to have that paper out before this paper is processed so that its results are available to strengthen the case argued in this paper.

5. On line 124, the authors state that there are two regions in the northern plains that show K, Th enrichments that are attributed to volcanism - it would be good to state why volcanism wouldn't be expected here in Eridania.

6. On line 127, the authors say: "That increases the spatial accuracy of the regions surrounding Eridania as the most HPE enriched Noachian terrain on Mars." What does this mean?

7. On line 128, the authors say: "The expectation that Al would resist leaching is realized with the presence of some high Al values in the area despite typically depleted values relative to the crust." - how do they tell this Al enrichment?

8. On line 176, the authors say they have previously shown the crustal heat flow results which are shown in Figure S2, but they don't give any idea of the equations or assumptions that went into this calculation. They cite an in press paper again but this isn't good enough and it seems more is required to explain how the curves were calculated. I think these assumptions are so important

they really should be in their own table.

9. On line 237, the authors say that volcanism should have led to viscoelastic relaxation, however they don't comment on the possibility of the lithosphere being strengthened by plume activity as in the heat pipe planets theory advanced here:

Moore, William B., Justin I. Simon, and A. Alexander G. Webb. "Heat-Pipe Planets." *Earth and Planetary Science Letters* 474 (September 15, 2017): 13–19.
<https://doi.org/10.1016/j.epsl.2017.06.015>.

10. On page 15, the authors do give some details I would expect to be earlier in the text, regarding their assumptions for their heat flow models - this level of specificity needs to be brought in to the discussions around line 176.

Finally, there a few problems with grammar that should also be easy to fix:

intro

"circulation, groundwater" should be "circulation, are groundwater"

"considerably" should be "relatively"

"than rocks rich" should be "compared to rocks rich"

"We show" should be "We shall show"

"Summarily" should be "In summary"

"U chemical" should be "Our U chemical"

"Overall that" should be "Overall this"

delete "rocks"

"Parala" should be "Paralana"

"area of Terra" should be "areas of Terra"

Best of luck,

Adrian Brown

Reviewer #2:

Remarks to the Author:

I enjoyed reading this paper arguing for the likelihood of long-term radiogenically heated hydrothermal system active on Mars' Eridania region. The arguments cover both observations and simple simulations that illustrate the likelihood of this scenario, and altogether provide a compelling story. This paper may help put such systems at the top of the list of places to explore with respect to our search for life outside Earth in the Solar System. The paper is a companion paper for a work in press with *Science Advances*, related to the effect of radiogenic heat under

thick ice cover.

I have only a few minor comments and suggestions:

L 102, please indicate the expected depth of Martian Moho.

L177-178, can you please repeat the assumption behind the $45\text{mW}\cdot\text{m}^{-2}$ value? I realise this is published, but given the importance of this statement, we need to grasp the assumptions and their limitations.

L184, "a deep hydrothermal setting where temperature likely exceeded 120C". This is an important statement, but strictly speaking it is not supported by reference [46], since the mineral assemblages include either prehnite or chlorite, with you do not list as part of the assemblages you consider. Please check whether this affects the estimated minimal temperatures. Since you do not need such high temperatures to sustain fluid flow, this is a minor comment, but it is important to get the geothermometry accurate in this paper.

Joel Brugger

Reviewer #3:

Remarks to the Author:

Overall the Ojha et al. manuscript entitled "Amagmatic hydrothermal systems on Mars from radiogenic heat" identifies a number of important issues but I recommend major revisions before publishing. The main argument outlined in the conclusion that the "lack of viscoelastic relaxation" in Eridania is an indicator of a long-lived radiogenic hydrothermal system lacks clarity and does not address major counter arguments effectively. Likewise the most important element of the paper in my opinion – the nature of the K and Th anomalies at Eridania receives less attention than it deserves and does not touch on some important points.

The main argument presented in the paper is somewhat difficult to follow and is never fully presented. The full argument seems to be that K and Th enrichments indicate the potential presence of a long-lived radiogenic heat source that substantially enhanced heat flow enough in the crust to power a long-lived hydrothermal system which resulted in substantial crustal cooling. This crustal cooling results in low/non-existent viscoelastic relaxation rates such that major impact craters in the region develop large mascons. Perhaps there is a reason this full argument is never presented all at once, because it sounds contradictory (and perhaps it is?). I don't mind so much, but some major points are missing in the discussion that should be addressed.

- Water budget – what are the hypothesized sources of water for this system and how are these squared with likely histories of the climate of the planet that don't provide easy recharge mechanisms? Certainly there seem to be large periods of time where recharge may be unlikely. Given the depth and breadth of this system, it might also result in massive storage of water in the crust which could be important for later studies on water budget and isotopic modeling. It would be interesting if this study could produce an overall water storage amount based on the degree of alteration it assumes.

- The arguments against magmatic intrusion don't seem to consider the possibility that it too could produce substantial hydrothermal cooling and thus create the mascons. The manuscript should explain how the hydrothermal cooling of a radiogenically driven hydrothermal system differs from that of a magmatically driven hydrothermal system? Certainly there is a goldilocks effect in play here – too much heat and you retain viscoelasticity in the crust, but too little heat and you don't get enough hydrothermal circulation? This tradeoff needs to be explained some what. Especially if you cool the crust too much – would you turn off the hydrothermal circulation?

The origin and nature of the K and Th anomalies and how they relate to the surface is really the interesting story here for me, and it would be good for the manuscript to delve into this more even if it comes at the expense of the long-lived hydrothermal system argument.

- Figure 2 is very difficult to understand and it is poorly explained. There seems to be a cutoff at higher latitudes which I understood to be the "mask" which is because of interference from H? But then there are weird areas of what appear to be dimness and brightness in the figure which are not well explained but may also be part of the mask? It is possible that the topography underneath the figure are lending apparent brightness enhancements to some parts and dimness to others which makes things confusing. I would rework this figure to make it more readable. Also please explain the presence of the triangle markers – are these the main basins?

- Since the manuscript concerns tying surface features to the K and Th abundances, it would be good to have some space devoted to the other two major zones of K and Th enrichment on Mars and the landforms they correspond to. It seems to be the whole vastitas borealis formation? Are there any connections between Eridania and the VBF?

- The manuscript should consider arguments presented in Irwin et al. 2013 which discusses the effects the ejecta from Hellas, Argyre, and Isidis. These three impact events coupled with major resurfacing from Tharsis and other major volcanic regions makes Eridania one of the few places that did not see major resurfacing since ~4 Ga and remains relatively dust free. Thus the high K and Th might simply be indicators of the ancient crust? Especially if this is something capture in the VBF which likely was emplaced very early on. Perhaps hydrothermalism was widespread on Early Mars and subsequent processes have buried it?

- In order to establish a baseline K and Th abundance for the crust, the manuscript uses a value of the bulk regolith. I think it would be useful to examine several different factors here including K and Th compositions derived from martian meteorites as well as from other major regions of the planet which seem homogeneous (VBF and major volcanic provinces of Syrtis and Hesperia?). Maybe rover results could also shed some light? Either way I think the paper would be improved with a broader discussion of K and Th abundances on Mars and how Eridania compares to other major regions.

- The paper mentions K/Th ratios being important indicators of aqueous alteration. Certainly the data do not have very good spatial resolution, but it would be interesting to see a K/Th ratio map of Eridania and the surrounding regions. There seem to be regions with Th enrichments that do not contain potassium. It would be interesting to see if these variations in K/Th ratio correspond with regions that show enhanced phyllosilicate formation or differ in other ways?

Overall I think this is a very compelling topic, and the authors have strong contributions to make here. There is already well established evidence for hydrothermalism in the region so acknowledging that radiogenic heat could have contributed to it is important, but the understanding of how Eridania fits with the global view that the K and Th maps provide is very interesting and I hope the authors consider these suggestions. My main concerns will be alleviated if they close up some of the holes in the arguments and address the points I brought up here with regard to the sources of the K and Th enrichments and the apparent self-contradictory nature of their argument.

Paul Niles

Response to Reviewers

KEY

BOLD: Reviewer's Query

Italicized: Our Response to the Reviewers

Italicized: Quotes from our Paper

Review of "Amagmatic hydrothermal systems on Mars from radiogenic heat" by Ojha et al.

This is a very nicely written paper that presents a study of the possibility of radiogenic heating causing hydrothermal systems to operate in the Noachian of Mars based on results from the Odyssey GRS data and focusing on the Eridania region.

Overall, I believe this is a very nice paper that could definitely be published in Nature Comms with some minor edits, mostly grammar related, which I will now go over.

We thank Dr. Brown for their key constructive, detailed, and prompt review. Please find our detailed response to your review below.

First, the minor scientific questions that could be dealt with quite quickly:

1. On Figure 2-6 and S1, I would recommend putting a label on the figure showing what the triangles stand for. I couldn't find this discussed in the text. The scale bars should be shorter for clarity and the longitude should be at the top of the figure, not the middle.

The black triangles show the location of the Eridania basins. We forgot to describe that in the figure legend, which is now fixed. The longitude is also now placed at the top of the figure for all maps. The color bar is the default shape and length in Matlab when placed at the bottom, so we could not really change its size. We did make the text bigger to make the images a bit clearer.

2. On line 40, the authors talk about hydrothermal activity influencing crustal magmatism, and they reference 2 articles, I would suggest they also reference this article:

Lillis, Robert J., Josef Dufek, Walter S. Kiefer, Benjamin A. Black, Michael Manga, Jacob A. Richardson, and Jacob E. Bleacher. "The Syrtis Major Volcano, Mars: A Multidisciplinary Approach to Interpreting Its Magmatic Evolution and Structural Development." *Journal of Geophysical Research: Planets* 120, no. 9 (September 1, 2015): 2014JE004774. <https://doi.org/10.1002/2014JE004774>.

We intentionally omitted references to some papers to abide by Nature Communications' policy of 70 references for articles (see <https://www.nature.com/ncomms/submit/article>).

At the time of submission, we had 99 references; thus, we could not cite more relevant work. Additionally, in the first paragraph of the paper, we only included work from Earth that illustrated hydrothermal circulation's potential effects in the biological and geological history.

Hence, why we end with the statement, "Hydrothermal systems may have played an equally important role in Mars' biological and geological history."

However, we discuss the role of hydrothermal activity on Mars's magnetic potential in the last paragraph before the discussion. We state, "Deep hydrothermal circulation and subsequent precipitation of single-domain magnetite could also have been a possible magnetization source in the Martian crust48,82,85." Here we provide references to three different works that have looked into the effect of hydrothermal activity on the Martian crustal magnetic field.

We looked into Lillis et al. (2015) and are uncertain whether the arguments presented in that paper are appropriate here. Specifically, Lillis et al. (2015) propose that the weak crustal magnetic fields over the Syrtis Major volcanic complex imply thermal demagnetization via magmatic intrusions. In contrast, our hypothesis for Terra Cimmeria and Terra Sirenum is that the crustal magnetic field could have been augmented by precipitation of magnetic minerals during hydrothermal activity.

An older paper by Lillis et al. (2008) looked into the possibility of the waning magnetic crustal magnetic susceptibility due to a decrease in hydrothermal alteration (see <https://doi.org/10.1029/2008GL034338>), which we cite in this revised version of the manuscript.

3. On line 50, the authors talk about the formation of minerals associated with hydrothermal activity, but did not mention carbonate as a possible serpentinization product, they probably should reference this study which looks at that possibility in Nili Fossae:

Brown, Adrian J., Christina E. Viviano, and Timothy A. Goudge. "Olivine-Carbonate Mineralogy of the Jezero Crater Region." *Journal of Geophysical Research: Planets* 125 (2020): <https://doi.org/10.1029/2019JE006011>.

We did overlook carbonates as potential products of serpentinization. Thank you for providing this reference, which we include in the updated version of the manuscript.

4. On line 98, the authors reference Ojha et al (in press), I think it would be good to have that paper out before this paper is processed so that its results are available to strengthen the case argued in this paper.

We agree. We inquired with the staff of Science Advances and they have set the publication date of that paper to be the first week of December. Thus, we are certain that the Science Advance paper will be published before this paper is out.

5. On line 124, the authors state that there are two regions in the northern plains that show K, Th enrichments that are attributed to volcanism - it would be good to state why volcanism wouldn't be expected here in Eridania.

We provide additional details on the K, Th enrichment in the two northern region and explain why the reason for K,Th enrichment in Eridania might be different. Text from the revised manuscript below:

“Outside Eridania, K and Th enrichment satisfying the enhanced Student’s-t test with spatially significant extent is limited to the northern lowlands, in the general area of the Vastitas Borealis Formation (VBF). Early works, by the Mars Odyssey mission team^{36,37}, characterized that extensively, and found a preferential association with increasing abundance of Surface Type 2 (ST2) mineralogy³⁸. Given a K/Th ratio corresponding to the crustal average, aqueous alteration (e.g., lowland water bodies) was excluded in preference for the volcanoclastic regolith derived from a mantle source with distinct large ion lithophile content^{36,39}. Using impact exposures of the shallow subsurface, previous study corroborated further that spatial correspondence with ST2 suggests Hesperian aged volcanoclastics⁶⁰, unrelated to the Noachian basement indicated by quasi-circular “ghost” craters of the lowlands. Specifically, mapped Noachian geology is absent in the K-Th enriched lowlands, characterized instead by 20 - 40 % areally of Hesperian volcanoclastics and up to ~10% Amazonian volcanoclastics [Fig 6 by Karunatillake et al., 2009]. That contrasts dramatically with the Eridania enrichment where > 60% of the area corresponds to Noachian units [cf., Fig 6 by Karunatillake et al., 2009]. Such association of the compositional signature with the younger upper regolith over the lowlands and VBF further separates the lowlands K-Th enrichment in time and provenance from Eridania’s.”

6. On line 127, the authors say: "That increases the spatial accuracy of the regions surrounding Eridania as the most HPE enriched Noachian terrain on Mars." What does this mean?

We provide further clarification on this statement as below:

“Furthermore, K and Th maps have the lowest spatial autocorrelation among all chemical maps as they are derived from gamma spectral features of natural radioactivity instead of galactic cosmic particle flux induced radioactivity that involves many layers of spectral modeling. The reduced spatial autocorrelation translates to high spatial accuracy unlike for other elements, ensuring^{46,57} that HPE enrichment is distinctive of Eridania in the Noachian terrain.”

7. On line 128, the authors say: "The expectation that Al would resist leaching is realized with the presence of some high Al values in the area despite typically depleted values relative to the crust." - how do they tell this Al enrichment?

We rely on the results shown in Figure 3 to show this and have modified the previous sentence as below:

“The expectation that Al would resist leaching is realized with the presence of some high Al values in the area despite typically depleted values relative to the crust as shown in Fig. 3.”

8. On line 176, the authors say they have previously shown the crustal heat flow results which are shown in Figure S2, but they don't give any idea of the equations or assumptions that went into this calculation. They cite an in press paper again but this isn't good enough and it seems more is

required to explain how the curves were calculated. I think these assumptions are so important they really should be in their own table.

Absolutely. Some of the assumptions that go into this estimate are already provided in the paper. For example, we describe the following assumptions in the paper:

“To assess if the observed concentration of K, Th, and U in Eridania and the surrounding region provided sufficient radiogenic heat for a hydrothermal system, we estimate plausible Noachian geotherms using heat flow estimates from our previous work⁸⁴. Crustal heat flow estimate depends on the heat production rate of the crust, crustal thickness, and crustal density (see Methodology). As in prior works, going back to those by Hahn et al. (2007, 2011), we approximate the bulk regolith chemistry to be representative of the crust⁸⁵.”

And,

“With these simplification, we estimated the crustal heat flow using GRS-derived chemical mass fraction maps of K and Th, gravity derived crustal thickness models, and estimate of crustal density from representative Martian meteorites⁸⁴.”

We do agree that some more details about the exact modeling should be provided in the paper. To that end, we have added the following section in the methodology to explain how we estimated the Noachian crustal heat flow using GRS data:

“The heat production rate of a crust (Q_c) is given by:

$$Q_c = \left[0.9928 C_U H_{238U} \exp\left(\frac{t \ln 2}{\tau_{238U}^{\frac{1}{2}}}\right) + 0.0071 C_U H_{235U} \exp\left(\frac{t \ln 2}{\tau_{235U}^{\frac{1}{2}}}\right) + C_{Th} H_{232Th} \exp\left(\frac{t \ln 2}{\tau_{232Th}^{\frac{1}{2}}}\right) + 1.191 \times 10^{-4} C_K H_{40K} \exp\left(\frac{t \ln 2}{\tau_{40K}^{\frac{1}{2}}}\right) \right]$$

(6)

Here, C and H represent the concentration and heat release constants of the radiogenic elements; t is time; and $\tau^{\frac{1}{2}}$ are the half-lives of the radioactive elements¹⁰¹. The concentration of the heat-producing elements is estimated using the GRS chemical maps. Heat release constants and the half-lives of the radiogenic elements are provided in ref (99). The heat production rate is multiplied by the estimate of crustal density and crustal thickness to derive a first-order estimate of Noachian crustal heat flow. We use gravity derived crustal thickness models which estimates the average Noachian crust to be >50 km. An average crustal density of 2900 kg m⁻³ is assumed for this work. The effect of these parameters on the average crustal heat flow is discussed in our previous work⁸⁴.”

9. On line 237, the authors say that volcanism should have led to viscoelastic relaxation, however they don't comment on the possibility of the lithosphere being strengthened by plume activity as in the heat pipe planets theory advanced here:

Moore, William B., Justin I. Simon, and A. Alexander G. Webb. “Heat-Pipe Planets.” *Earth and Planetary Science Letters* 474 (September 15, 2017): 13–19. <https://doi.org/10.1016/j.epsl.2017.06.015>.

Absolutely. This is certainly a possibility; however, heat-pipe related processes operate on a global scale and should have led to the preservation of the Moho topography of impact basins elsewhere too. As shown in Figure 5, only a handful of Noachian aged impact basins show noticeable positive Bouguer anomaly. The limited numbers of craters that seem to have their Moho preserved instead argues for a spatially limited process (such as local hydrothermal circulation) that could be responsible for their preservation. We add the following lines to convey this in the paper:

“The preservation of ancient gravity anomalies at Newton and other basins of Eridania could also be due to heat-pipe driven process which operates on a global scale and is postulated to produce a thick, cold, and strong lithosphere very early in Mars’ history⁸⁶. However, such a process should have led to the widespread preservation of the Moho topography of impact basins, which is not observed (Fig.5).”

10. On page 15, the authors do give some details I would expect to be earlier in the text, regarding their assumptions for their heat flow models - this level of specificity needs to be bought in to the discussions around line 176.

Right. Since the Methodology section is placed at the end in Nature Communication format, the difficulty is not to saturate the results section with technical details about the methodology. In this version, we add key assumptions about the work in the main text while alerting readers to refer to Methodology for more details on the technical aspect of this work. For example, in the results section, we now state the following:

“Crustal heat flow estimate depends on the heat production rate of the crust, crustal thickness, and crustal density (see Methodology).”

The methods section itself gives a detailed explanation of how we calculated the heat flow. We have added more specificity of our methods in the results section in this version. See our response to your earlier comment above.

Finally, there a few problems with grammar that should also be easy to fix:

Thanks for catching these. Without a line number reference, it was a bit confusing to associate your comments to the text, but we think we fixed them all.

intro

"circulation, groundwater" should be "circulation, are groundwater"
Thanks. Fixed.

"considerably" should be "relatively"
Thanks. Fixed.

"than rocks rich" should be "compared to rocks rich"

Thanks. Fixed.

"We show" should be "We shall show"

Thanks. Fixed.

"Summarily" should be "In summary"

Thanks. Fixed.

"U chemical" should be "Our U chemical"

Thanks. Fixed.

"Overall that" should be "Overall this"

Thanks. Fixed.

delete "rocks"

Thanks. Fixed.

"Parala" should be "Paralana"

Thanks. Fixed.

"area of Terra" should be "areas of Terra"

Thanks. Fixed.

Best of luck,

Adrian Brown

Reviewer #2 (Remarks to the Author):

I enjoyed reading this paper arguing for the likelihood of long-term radiogenically heated hydrothermal system active on Mars' Eridania region. The arguments cover both observations and simple simulations that illustrate the likelihood of this scenario, and altogether provide a compelling story. This paper may help put such systems at the top of the list of places to explore with respect to our search for life outside Earth in the Solar System. The paper is a companion paper for a work in press with Science Advances, related to the effect of radiogenic heat under thick ice cover.

We thank Dr. Brugger for their prompts and constructive comments. We are also appreciative of your key work on radiogenic hydrothermal systems on Earth. Identification of potential radiogenic hydrothermal systems on other worlds would not have been possible without terrestrial perspectives. Please find our detailed response to your reviews below: .

I have only a few minor comments and suggestions:

L 102, please indicate the expected depth of Martian Moho.

Based on the gravity derived crustal thickness models of Mars, the expected depth to the Martian Moho can vary considerably based on the location. The average depth to the Moho in the Southern highlands of Mars is estimated to be ~50 km.

In our crater relaxation simulation, we model a crust of 50 km thickness with a mantle uplift at the center of the crater. The amplitude of the mantle uplift underneath the crater is shown in Figure 7 and is approximately 25 km.

Instead of adding this detail in the introduction section, we have added this information in the methodology:

“The crustal thickness in our simulation is set to 50 km, equal to that of the average crustal thickness of Mars⁹⁵.”

“After an impact event, the mantle beneath the crater depression is uplifted to the isostatic equilibrium level (e.g., Namiki et al 2009), the amplitude (h) of which can be approximated by a simple mass balance equation:

$$h = \frac{\rho_c}{(\rho_m - \rho_c)} z;$$

where ρ_c and ρ_m are the densities of the crust and mantle, and z is the depth of the crater. Assuming a crustal density of 2900 kg m⁻³, a mantle density of 3500 kg m⁻³, Newton crater's current depth of 3500 m, the mantle uplift likely exceeded 20 km at the time of the formation of this basin. We model the shape of the mantle to resemble a gaussian like function similar to our previous work⁷². The total number of elements in the mesh exceeds 10⁷.”

L177-178, can you please repeat the assumption behind the 45mW • m⁻² value? I realise this is

published, but given the importance of the this statement, we need to grasp the assumptions and their limitations.

Absolutely. We have added the following in the methods section to explain the reasoning behind the 45 mW m² value.

Crustal Heat Flow Estimates

“The heat production rate of a crust (Q_c) is given by:

$$Q_c = \left[\frac{0.9928 C_U H_{238U} \exp\left(\frac{t \ln 2}{\tau_{238U}^{\frac{1}{2}}}\right) + 0.0071 C_U H_{235U} \exp\left(\frac{t \ln 2}{\tau_{235U}^{\frac{1}{2}}}\right) + C_{Th} H_{232Th} \exp\left(\frac{t \ln 2}{\tau_{232Th}^{\frac{1}{2}}}\right) + 1.191 \times 10^{-4} C_K H_{40K} \exp\left(\frac{t \ln 2}{\tau_{40K}^{\frac{1}{2}}}\right) \right]$$

(6)

Here, C and H represent the concentration and heat release constants of the radiogenic elements; t is time; and $\tau^{\frac{1}{2}}$ are the half-lives of the radioactive elements⁹¹. The concentration of the heat-producing elements is estimated using the GRS chemical maps. Heat release constants and the half-lives of the radiogenic elements are provided in ref (99). The heat production rate is multiplied by the estimate of crustal density and crustal thickness to derive a first-order estimate of Noachian crustal heat flow. We use gravity derived crustal thickness models which estimates the average Noachian crust to be >50 km. An average crustal density of 2900 kg m³ is assumed for this work. The effect of these parameters on the average crustal heat flow is discussed in our previous work⁸¹.”

In the main text, we provide a brief description on heat flow calculation and refer the readers to the Methods section for more details:

“Crustal heat flow estimate depends on the heat production rate of the crust, crustal thickness, and crustal density (see Methodology).”

And

“With these simplification, we estimated the crustal heat flow using GRS-derived chemical mass fraction maps of K and Th, gravity derived crustal thickness models, and estimate of crustal density from representative Martian meteorites⁸¹.”

L184, "a deep hydrothermal setting where temperature likely exceeded 120C". This is an important statement, but strictly speaking it is not supported by reference [46], since the mineral assemblages include either prehnite or chlorite, with you do not list as part of the assemblages you consider. Please check wether this affects the estimated minimal temperatures. Since you do not need such high temperatures to sustain fluid flow, this is a minor comment, but it is important to get the geothermometry accurate in this paper.

We agree and upon double checking ref(46) we have revised this line to say the following:

“The various hydrothermal phases identified in the Eridania region, such as saponite, talc-saponite, Fe-rich mica, Fe- and Mg-serpentine, Mg-Fe-Ca carbonate, and probable Fe-sulphide²¹ implicates a formation environment in a deep hydrothermal setting where fluids with elevated temperature

interacted with the host rocks⁴⁷. For example, serpentines form by hydrothermal alteration of ultramafic rocks at temperatures ranging from ambient to 400 °C⁵. Nontronite formation on Mars requires elevated temperature in the range 20 - 40 °C⁷. Carbonates. A steady-state geotherm with our heat flow estimates and a mean crustal conductivity of 3 W m⁻¹ K⁻¹ show that these temperature values could have easily exceeded at shallow depth in Eridania (Fig. S2). Thus, the observed K, Th, and computed U abundance in the Eridania region could have readily sustained a radiogenic heat-driven hydrothermal system.”

Joel Brugger

Reviewer #3 (Remarks to the Author):

Overall the Ojha et al. manuscript entitled “Amagmatic hydrothermal systems on Mars from radiogenic heat” identifies a number of important issues but I recommend major revisions before publishing. The main argument outlined in the conclusion that the “lack of viscoelastic relaxation” in Eridania is an indicator of a long-lived radiogenic hydrothermal system lacks clarity and does not address major counter arguments effectively. Likewise the most important element of the paper in my opinion – the nature of the K and Th anomalies at Eridania receives less attention than it deserves and does not touch on some important points.

We thank Dr. Niles for their prompt and constructive reviews. We have restructured our argument about the lack of viscoelastic relaxation and how it supports the notion of a long-lived hydrothermal system in this version of the manuscript. We also devote more text to the nature of the K and Th anomalies at Eridania and other regions of Mars. Please find our detailed response to your review below:

The main argument presented in the paper is somewhat difficult to follow and is never fully presented. The full argument seems to be that K and Th enrichments indicate the potential presence of a long-lived radiogenic heat source that substantially enhanced heat flow enough in the crust to power a long-lived hydrothermal system which resulted in substantial crustal cooling. This crustal cooling results in low/non-existent viscoelastic relaxation rates such that major impact craters in the region develop large mascons. Perhaps there is a reason this full argument is never presented all at once, because it sounds contradictory (and perhaps it is?). I don't mind so much, but some major points are missing in the discussion that should be addressed.

Your summary of our main argument is correct. However, our intention in the paper was not to make the argument that these basins (Newton, Copernicus, and Eridania basins) are mascons. A diagnostic characteristic of a mascon is a positive free-air gravity anomaly. None of the Eridania basins, Newton, or Copernicus crater are have a positive free-air gravity anomaly, so none of them are mascons. We have added a new figure (Fig. S6) that shows the Free-air gravity of Mars to illustrate this.

This is something that was clearly unclear in the paper, so we have added the following line to explicitly state that none of the basins in the Eridania region are mascons.

“A diagnostic characteristic of a mascon is a positive free-air gravity anomaly which Newton, Copernicus, and Eridania basins lack.”

We clarified and elaborated on the introduction section to present our main argument:

“Michalski et al. (2017) considered magmatic intrusion as a possible heat source that sustained the Eridania hydrothermal system. However, magmatic intrusions would be considerably short-lived compared to rocks rich in HPE sustaining the hydrothermal system in Eridania. Here we show that Eridania and the surrounding regions have the highest abundance of Th, K, and cosmochemically equivalent U of any Noachian terrain. By combining chemical maps with geophysical data of Mars, we have shown that the region surrounding Eridania had one of the highest crustal heat flows on

Mars during the Noachian (Ojha et al., In Press). Using this prior heat flow estimate, we show that the shallow subsurface temperature at Eridania would have easily exceeded the temperature threshold required for hydrothermal systems. In the Eridania region, several large impact basins have prominent Bouguer gravity anomalies suggestive of lack of viscoelastic relaxation of the Moho topography⁴⁸. We show that the lack of viscoelastic relaxation of the Moho topography in this region is likely due to the crustal cooling by hydrothermal circulation, similar to the cooling of the oceanic and continental lithosphere by hydrothermal circulation on Earth^{4,8,9}. The Eridania basin is located within the broader Terra Cimmeria/ Terra Sirenum region, which has one of the highest crustal magnetic fields on Mars. Such a high crustal magnetic field on Mars may also reflect augmentation of the primordial natural remnant magnetization by a hydrothermally induced chemical remnant magnetic field^{10,50}. In summary, via synthesis of available geophysical and geochemical data of Mars along with numerical models of crater relaxation, we show that the Eridania region on Mars was potentially a long-lived radiogenic hydrothermal system that may have contributed to the region's enigmatic thermal and magnetic history besides providing a long-lived habitable environment."

- Water budget - what are the hypothesized sources of water for this system and how are these squared with likely histories of the climate of the planet that don't provide easy recharge mechanisms? Certainly there seem to be large periods of time where recharge may be unlikely. Given the depth and breadth of this system, it might also result in massive storage of water in the crust which could be important for later studies on water budget and isotopic modeling. It would be interesting if this study could produce an overall water storage amount based on the degree of alteration it assumes.

The source of water for this system is not well understood, but the concave hypsometry of the basins suggest that these basins were likely covered by a thick ice cover (e.g., Michalski et al., 2017). If so, the provenance of the liquid water is likely from the basal melting of the ice sheets which is something we explore in a great deal in our companion paper in Science Advances. We have added the following lines in the introduction about the origin of the water in Eridania basins:

"The provenance of the liquid water in the Eridania basins is not entirely understood. The relatively sparse valley networks in Eridania does not support surface runoff as the dominant water source⁹¹. Instead, the scarp-bounded benches in Gorgonum chaos⁹², the concave hypsometry of Eridania basins^{21,46}, and the reducing deep-water environment indicated by Fe²⁺ - rich clay minerals is suggestive of Eridania being an ice-covered sea."

We think that the estimation of an overall water storage amount based on the degree of alteration is beyond the scope of this work. The groundwater storage potential would depend on the porosity profile of the shallow crust. In our companion paper, we explore this topic a little bit, however, for this paper we think an estimation of overall water storage amount does not necessarily help corroborate the main thesis of the paper, which is that the Eridania hydrothermal system could have been powered by radiogenic decay of HPE.

- The arguments against magmatic intrusion don't seem to consider the possibility that it too could produce substantial hydrothermal cooling and thus create the mascons. The manuscript should explain how the hydrothermal cooling of a radiogenically driven hydrothermal system differs from

that of a magmatically driven hydrothermal system? Certainly there is a goldilocks effect in play here – too much heat and you retain viscoelasticity in the crust, but too little heat and you don’t get enough hydrothermal circulation? This tradeoff needs to be explained some what. Especially if you cool the crust too much – would you turn off the hydrothermal circulation?

Our writing style and lack of illustrations may have obscured what we intended. A diagnostic characteristic of a mascon is a positive free-air gravity anomaly, which is not observed at Eridania basins, Newton, or Copernicus crater. Thus, neither of these basins are mascons. We first clarify this point in the results section as follow and have included a new figure in the SOM that shows the Free-air gravity map of Mars.

“However, none of the basins considered here Newton, Copernicus, or Eridania basins are mascons, since they lack a positive free-air gravity anomaly (Fig. S6).”

Figure S6. Free-air gravity anomaly map of Mars. The circles show impact craters on Mars with a diameter between 200 and 500 km. Impact craters with associated Bouguer anomaly are shown in black while white circles show impact craters with no prominent Bouguer anomaly (see Fig. 5). The triangles show all identified volcanic structures on Mars.

As you can see from the Free-air gravity map of Mars none of these basins have positive Free-air gravity values. We then further clarify in this revised manuscript why one would expect viscoelastic relaxation with magmatic intrusion but not with radiogenic heat in the following way:

“If the Eridania hydrothermal system were primarily driven by the heat from magmatic intrusion at great depths” instead of heat from radioactivity in the crust, then the high basal heat flow should have induced significant viscoelastic relaxation of the surface and Moho topography in this region. While both heat sources can elevate the temperature in the crust and drive hydrothermal circulation, radioactivity within the crust does not notably change the temperature structure at depths (Fig. S3). In contrast, the temperature at depth is significantly augmented in the presence of

a magmatic heat source (Fig. S3). Thus, if the hydrothermal system in Eridania was driven by magmatic heat at depth, we should expect the basins to have undergone viscoelastic relaxation.”

The origin and nature of the K and Th anomalies and how they relate to the surface is really the interesting story here for me, and it would be good for the manuscript to delve into this more even if it comes at the expense of the long-lived hydrothermal system argument.

- Figure 2 is very difficult to understand and it is poorly explained. There seems to be a cutoff at higher latitudes which I understood to be the “mask” which is because of interference from H? But then there are weird areas of what appear to be dimness and brightness in the figure which are not well explained but may also be part of the mask? It is possible that the topography underneath the figure are lending apparent brightness enhancements to some parts and dimness to others which makes things confusing. I would rework this figure to make it more readable. Also please explain the presence of the triangle markers – are these the main basins?

*We appreciate the comments. Indeed, the cutoff at higher latitudes is due to interference from H. The legend of the figure explains this:
“Rapidly increasing H abundance dilutes and increases numerical uncertainty for HPE concentrations in the polar latitudes. A mask has thus been applied to exclude such areas.”*

The dimness and brightness of the figure is an “artifact” of overlaying GRS map on top of MOLA shaded relief. Ojha uses Apple OS and does not have access to ArcMap. The overlay figure was made using a custom Matlab code which is not aesthetically pleasing as shaded figures created using ArcMap or some other GIS platform. In order to avoid any misunderstanding with the figures, we have removed the MOLA shaded relief from the background. The map now only shows the GRS chemical maps and those apparent brightness and dimness is now gone. Hopefully, the figure is much more readable now.

*Thanks for noticing the missing explanation about the triangles. We have added the following line to explain that the triangles show the Eridania basins:
“The black triangles show the location of Eridania basins.”*

- Since the manuscript concerns tying surface features to the K and Th abundances, it would be good to have some space devoted to the other two major zones of K and Th enrichment on Mars and the landforms they correspond to. It seems to be the whole vastitas borealis formation? Are there any connections between Eridania and the VBF?

We have added a paragraph dedicated to the discussion about the other two major zones of K and Th enrichment.

“Outside Eridania, K and Th enrichment satisfying the enhanced Student’s-t test with spatially significant extent is limited to the northern lowlands, in the general area of the Vastitas Borealis Formation (VBF). Early works, by the Mars Odyssey mission team^{36,37}, characterized that extensively, and found a preferential association with increasing abundance of Surface Type 2 (ST2) mineralogy³⁸. Given a K/Th ratio corresponding to the crustal average, aqueous alteration (e.g., lowland water bodies) was excluded in preference for the volcanoclastic regolith derived from a mantle source with distinct large ion lithophile content^{36,39}. Using impact exposures of the shallow

subsurface, previous study corroborated further that spatial correspondence with ST2 suggests Hesperian aged volcanoclastics⁶⁰, unrelated to the Noachian basement indicated by quasi-circular “ghost” craters of the lowlands. Specifically, mapped Noachian geology is absent in the K-Th enriched lowlands, characterized instead by 20 - 40 % areally of Hesperian volcanoclastics and up to ~10% Amazonian volcanoclastics [Fig 6 by Karunatillake et al., 2009]. That contrasts dramatically with the Eridania enrichment where > 60% of the area corresponds to Noachian units [cf., Fig 6 by Karunatillake et al., 2009]. Such association of the compositional signature with the younger upper regolith over the lowlands and VBF further separates the lowlands K-Th enrichment in time and provenance from Eridania’s.”

- The manuscript should consider arguments presented in Irwin et al. 2013 which discusses the effects the ejecta from Hellas, Argyre, and Isidis. These three impact events coupled with major resurfacing from Tharsis and other major volcanic regions makes Eridania one of the few places that did not see major resurfacing since ~4 Ga and remains relatively dust free. Thus the high K and Th might simply be indicators of the ancient crust? Especially if this is something capture in the VBF which likely was emplaced very early on. Perhaps hydrothermalism was widespread on Early Mars and subsequent processes have buried it?

Prior works [e.g., Baratoux et al., 2011] suggest that K and Th are generally enriched in younger volcanism, which diverges from the possibility of an ancient crust enriched in large ion lithophiles. Nevertheless, this is a great point that we had not considered. We added the following lines to highlight the fact that Eridania may be one of the few places that did not see major resurfacing from large impacts:

“Further, ejecta from Hellas is expected to only thinly mantle the highlands of Terra Cimmeria and Terra Sirenum; thus, the Eridania surrounding region may represent the pre-Hellas surface better than most other regions on Mars⁷².”

- In order to establish a baseline K and Th abundance for the crust, the manuscript uses a value of the bulk regolith. I think it would be useful to examine several different factors here including K and Th compositions derived from martian meteorites as well as from other major regions of the planet which seem homogeneous (VBF and major volcanic provinces of Syrtis and Hesperia?). Maybe rover results could also shed some light? Either way I think the paper would be improved with a broader discussion of K and Th abundances on Mars and how Eridania compares to other major regions.

We now include the detailed context of K and Th enrichment in Eridania compared to the only other regions of enrichment on the planet. In situ data do not include Th, and none of the sites are proximal to or correspond in mapped geology with Eridania. So we avoided in situ data to avoid speculative misinterpretations. Likewise, meteorites, given their ill-defined provenance locales on Mars, may precipitate misinterpretations for Eridania despite their utility for planetary scale processes [cf., Taylor et al., 2006] as in the work by Balta and McSween [2013].

- The paper mentions K/Th ratios being important indicators of aqueous alteration. Certainly the data do not have very good spatial resolution, but it would be interesting to see a K/Th ratio map

of Eridania and the surrounding regions. There seem to be regions with Th enrichments that do not contain potassium. It would be interesting to see if these variations in K/Th ratio correspond with regions that show enhanced phyllosilicate formation or differ in other ways?

We have added the following sentence to clarify that the global K/Th variability has already been considered in the literature, which would make added work by us redundant: “Equally important, hydrothermalism is known to leach K^{763-67} , decreasing the residual rock’s K/Th, which we observe, albeit not at the level of anomalies found in regions like Memnonia Fossae [cf., Taylor et al., 2006].”

Mineral detections, unless we establish their lateral extent by means of stratigraphic continuity beneath the landscape, would open more room for misinterpretations given the dramatic contrast in lateral resolution between nuclear and VNIR spectroscopy. That being said, we did make a new map that shows the spatial distribution of phyllosilicates on top of the K/Th map. In general, we do observe that regions with large phyllosilicate concentration have lower K/Th values.

We include new Figure (Fig. S1) that shows this putative relationship between K/Th ratio and phyllosilicate distribution and have added the following lines in the main text:

“The distribution of phyllosilicates also shows a putative correlation with regions with a low K/Th ratio on Mars; however, this interpretation may be misleading due to the dramatic contrast in lateral resolution between nuclear and VNIR spectroscopy.”

Overall I think this is a very compelling topic, and the authors have strong contributions to make here. There is already well established evidence for hydrothermalism in the region so acknowledging that radiogenic heat could have contributed to it is important, but the understanding of how Eridania fits with the global view that the K and Th maps provide is very interesting and I hope the authors consider these suggestions.

Thank you again for this comprehensive review of our manuscript. Your revisions have helped significantly improve the clarity and the scope of the paper.

My main concerns will be alleviated if they close up some of the holes in the arguments and address the points I brought up here with regard to the sources of the K and Th enrichments and the apparent self-contradictory nature of their argument.

We hope that the added description about the relationship of the viscoelastic relaxation of the Moho in relation to various heat sources clear any apparent self-contradictory nature of our argument.

Paul Niles

Reviewers' Comments:

Reviewer #1:

Remarks to the Author:

I congratulate the authors, they have dealt with all my concerns and I recommend the paper be published, however with one proviso, I would like to see the Science Advances paper come out so it can be referenced - I will leave it to the editor to deal with this concern. Apparently it should be coming out this week.

Also, a minor thing, but the equations will need to be renumbered with the new equation being added.

Best of luck,

Adrian Brown

Reviewer #2:

Remarks to the Author:

The revised version of the ms has made some important clarifications, taking into account the comments of the three reviewers.

In my view, the only part that requires a bit of work in Figure 2c. The legend needs to be clarified: red lines are t-values for T_h , and black ones for K , I assume? Please give full name of the 't' parameter in the legend: "enhanced Student's t-test parameter, t_i " (same notation as in the method section). What is the grey-scale map in the background (and grey-scale legend as appropriate).

Finally, the t-contours look terrible and the labels are, in part, impossible to read. Please edit the figure for clarity. This is important as the absolute values of the t-parameters are actually discussed in the text (NB matlab contour maps should be editable if saved as eps).

Joel Brugger

Reviewer #3:

Remarks to the Author:

My thanks to the authors for their excellent responses and clarifications. I find the revised text and figures satisfy my concerns and I'm happy to recommend this for publication. Clearly I was misusing the term "mascons" in my review and I apologize for any misunderstanding.

Paul Niles

Response to Reviewers - II

KEY

BOLD: Reviewer's Query

Italicized: Our Response to the Reviewers

Italicized: Quotes from our Paper

Reviewer #1 (Remarks to the Author):

I congratulate the authors, they have dealt with all my concerns and I recommend the paper be published, however with one proviso, I would like to see the Science Advances paper come out so it can be referenced - I will leave it to the editor to deal with this concern. Apparently it should be coming out this week.

Also, a minor thing, but the equations will need to be renumbered with the new equation being added.

Best of luck,

Adrian Brown

Thank you again Dr. Brown for your constructive and detailed review. The Science Advance paper is now published and cited in the paper. The equations are also renumbered.

Reviewer #2 (Remarks to the Author):

The revised version of the ms has made some important clarifications, taking into account the comments of the three reviewers.

In my view, the only part that requires a bit of work in Figure 2c. The legend needs to be clarified: red lines are t-values for T_h , and black ones for K , I assume? Please give full name of the 't' parameter in the legend: "enhanced Student's t-test parameter, t_i " (same notation as in the method section). What is the grey-scale map in the background (and grey-scale legend as appropriate).

Finally, the t-contours look terrible and the labels are, in part, impossible to read. Please edit the figure for clarity. This is important as the absolute values of the t-parameters are actually discussed in the text (NB matlab contour maps should be editable if saved as eps).

Joel Brugger

Thank you again Dr. Brugger for your constructive and detailed review. We have revised Figure 2 (c) to make it clearer.

Reviewer #3 (Remarks to the Author):

My thanks to the authors for their excellent responses and clarifications. I find the revised text and figures satisfy my concerns and I'm happy to recommend this for publication. Clearly I was misusing the term "mascons" in my review and I apologize for any misunderstanding.

Paul Niles

Thank you again Dr. Niles for your constructive and detailed review.